# Self-supervised and Supervised Joint Training for Resource-rich Machine Translation

## Abstract

Self-supervised pre-training of text representations has been successfully applied to low-resource Neural Machine Translation (NMT). However, it usually fails to achieve notable gains on resource-rich NMT. In this paper, we propose a joint training approach, $F_2$-*XEnDec*, to combine self-supervised and supervised learning to optimize NMT models. To exploit complementary self-supervised signals for supervised learning, NMT models are trained on examples that are interbred from monolingual and parallel sentences through a new process called crossover encoder-decoder. Experiments on two resource-rich translation benchmarks, WMT'14 English-German and WMT'14 English-French, demonstrate that our approach achieves substantial improvements over several strong baseline methods and obtains a new state of the art of 46 BLEU on English-French when incorporating back translation. Results also show that our approach is capable of improving model robustness to input perturbations, particularly to code-switching noise which usually appears on the social media.

## 1 Introduction

Self-supervised pre-training of text representations (Peters et al., 2018; Radford et al., 2018) has achieved tremendous success in natural language processing applications. Inspired by BERT (Devlin et al., 2019), recent works attempt to leverage sequence-to-sequence model pre-training for Neural Machine Translation (NMT). Generally, these methods comprise two stages: pre-training and finetuning. During the pre-training stage, a proxy task, *e.g.* the Cloze task (Devlin et al., 2019), is used to learn the model parameters on abundant unlabeled monolingual data. In the second stage, the full or partial model is finetuned on a downstream translation task of labeled parallel sentences. When the amount of labeled data is limited, studies have demonstrated the benefit of pre-training for low-resource translation tasks (Lewis et al., 2019; Song et al., 2019).

In many NMT applications, we are confronted with resource-rich languages which are characterized by millions of labeled parallel sentences. However, for these resource-rich tasks, pre-training representations rarely endows the NMT model with superior quality and, even worse, it sometimes can undermine the model's performance if improperly utilized (Zhu et al., 2020). This is partly due to catastrophic forgetting (French, 1999) where prolonged finetuning on large corpora causes the learning to overwhelm the knowledge learned during pre-training. Several mitigation methods have been proposed for resource-rich machine translation (Edunov et al., 2019; Yang et al., 2019; Zhu et al., 2020), such as freezing the pre-trained representations during the finetuning stage.

In this paper, we study resource-rich machine translation through a different perspective of joint training where in contrast to the conventional two-stage approaches, we train NMT models in a single stage using the self-supervised objective (on unlabeled monolingual sentences) in addition to the supervised objective (on labeled parallel sentences). The biggest challenge for this single-stage training paradigm is that self-supervised learning is less useful in joint training because it provides a much weaker learning signal that is easily dominated by the signal obtained through supervised learning. As a result, plausible approaches such as simply combining self-supervised and supervised learning objectives perform not much better than the supervised learning objective by itself.

To this end, we introduce an approach to exploit complementary self-supervised learning signals to facilitate supervised learning in a joint training framework. Inspired by chromosomal

crossovers (Rieger et al., 2012), we propose a new task called crossover encoder-decoder (or *XEn-Dec*) which takes two training examples as inputs (called parents), shuffles their source sentences, and produces a "virtual" sentence (called offspring) by a mixture decoder model. The key to our approach is to "interbreed" monolingual (unlabeled) and parallel (labeled) sentences through *second filial generation* with a crossover encoder-decoder, which we call $F_2$-*XEnDec*, and train NMT models on the $F_2$ offspring. As the $F_2$ offspring exhibits combinations of traits that differ from those found in either parent, it turns out to be a meaningful objective to learn NMT models from both labeled and unlabeled sentences in a joint training framework.

To the best of our knowledge, the proposed $F_2$-*XEnDec* is among the first joint training approaches that substantially improve resource-rich machine translation. Closest to our work are two-stage approaches by Zhu et al. (2020) and Yang et al. (2019) who designed special finetuning objectives. Compared to their approaches, our focus lies on addressing a different challenge which is *making self-supervised learning complementary to joint training of supervised NMT models on large labeled parallel corpora*. Our experimental results substantiate the competitiveness of the proposed joint training approach. Furthermore, our results suggest that the approach improves the robustness of NMT models (Belinkov & Bisk, 2018; Cheng et al., 2019). Contemporary NMT systems often lack robustness and therefore suffer from dramatic performance drops when they are exposed to input perturbations, even though these perturbations may not be strong enough to alter the meaning of the input sentence. Our improvement in robustness is interesting as none of the two-stage training approaches have ever reported this behavior.

We empirically validate our approach on the WMT'14 English-German and WMT'14 English-French translation benchmarks which yields an improvement of 2.13 and 1.78 BLEU points over a vanilla Transformer model baseline. We also achieve a new state of the art of 46 BLEU on the WMT'14 English-French translation task when further incorporating the back translation technique into our approach. In summary, our contributions are as follows:

1. We propose a crossover encoder-decoder (*XEnDec*) that generates "virtual" examples over pairs of training examples. We discuss its relation to the standard self-supervised learning objective that can be recovered by *XEnDec*.

2. We combine self-supervised and supervised losses in a joint training framework using our proposed $F_2$-*XEnDec* and show that self-supervised learning is complementary to supervised learning for resource-rich NMT.

3. Our approach achieves significant improvements on resource-rich translation tasks and exhibits higher robustness against input perturbations, particularly to code-switching noise.

## 2 BACKGROUND

### 2.1 NEURAL MACHINE TRANSLATION

Under the encoder-decoder paradigm (Bahdanau et al., 2015; Gehring et al., 2017; Vaswani et al., 2017), the conditional probability $P(\boldsymbol{y}|\boldsymbol{x};\boldsymbol{\theta})$ of a target-language sentence $\boldsymbol{y} = y_1, \cdots, y_J$ given a source-language sentence $\boldsymbol{x} = x_1, \cdots, x_I$ is modeled as follows: The encoder maps the source sentence $\boldsymbol{x}$ onto a sequence of $I$ word embeddings $e(\boldsymbol{x}) = e(x_1), ..., e(x_I)$. Then the word embeddings are encoded into their corresponding continuous hidden representations. The decoder acts as a conditional language model that reads embeddings $e(\boldsymbol{y})$ for a shifted copy of $\boldsymbol{y}$ along with the aggregated contextual representations $\boldsymbol{c}$. For clarity, we denote the input and output in the decoder as $\boldsymbol{z}$ and $\boldsymbol{y}$, *i.e.* $\boldsymbol{z} = \langle s \rangle, y_1, \cdots, y_{J-1}$, where $\langle s \rangle$ is a start symbol. Conditioned on the aggregated contextual representation $\boldsymbol{c}_j$ and its partial target input $\boldsymbol{z}_{\leq j}$, the decoder generates $\boldsymbol{y}$ as $P(\boldsymbol{y}|\boldsymbol{x};\boldsymbol{\theta}) = \prod_{j=1}^{J} P(y_j|\boldsymbol{z}_{\leq j}, \boldsymbol{c}_j; \boldsymbol{\theta})$. The aggregated contextual representation $\boldsymbol{c}$ is often calculated by summarizing the sentence $\boldsymbol{x}$ with an attention mechanism (Bahdanau et al., 2015). A byproduct of the attention computation is a noisy alignment matrix $\boldsymbol{A} \in \mathbb{R}^{J \times I}$ which roughly captures the translation correspondence between target and source words (Garg et al., 2019).

Generally, NMT optimizes the model parameters $\boldsymbol{\theta}$ by minimizing the empirical risk over a parallel training set $(\boldsymbol{x}, \boldsymbol{y}) \in \mathcal{S}$:

$$\mathcal{L}_{\mathcal{S}}(\boldsymbol{\theta}) = \mathop{\mathbb{E}}_{(\boldsymbol{x},\boldsymbol{y}) \in \mathcal{S}} [\ell(f(\boldsymbol{x}, \boldsymbol{y}; \boldsymbol{\theta}), h(\boldsymbol{y}))], \tag{1}$$

where $\ell$ is the cross entropy loss between the model prediction $f(\boldsymbol{x}, \boldsymbol{y}; \boldsymbol{\theta})$ and $h(\boldsymbol{y})$, and $h(\boldsymbol{y})$ denotes the sequence of one-hot label vectors for $\boldsymbol{y}$ with label smoothing in the Transformer (Vaswani et al., 2017).

## 2.2 Pre-training for Neural Machine Translation

Pre-training sequence-to-sequence models for language generation is receiving increasing attention in the machine translation community (Song et al., 2019; Lewis et al., 2019). These methods generally comprise two stages: pre-training and finetuning. The pre-training takes advantage of the abundant monolingual corpus $\mathcal{U} = \{\boldsymbol{y}\}$ to learn representations through a self-supervised objective called denoising autoencoder (Vincent et al., 2008). The denoising autoencoder aims at reconstructing the original sentence $\boldsymbol{y}$ from one of its corrupted counterparts. Let $\boldsymbol{y}^\diamond$ be obtained by corrupting $\boldsymbol{y}$ with a noise function $n(\cdot)$ and masking words.

Then the pseudo parallel data $(\boldsymbol{y}^\diamond, \boldsymbol{y})$ is fed into the NMT model to compute the reconstruction loss. The self-supervised loss over the monolingual corpus $\mathcal{U}$ is defined as:

$$\mathcal{L}_{\mathcal{U}}(\boldsymbol{\theta}) = \mathbb{E}_{\boldsymbol{y} \in \mathcal{U}}[\ell(f(\boldsymbol{y}^\diamond, \boldsymbol{y}; \boldsymbol{\theta}), h(\boldsymbol{y}))], \tag{2}$$

The optimal model parameters $\boldsymbol{\theta}^\star$ are learned via a self-supervised loss $\mathcal{L}_{\mathcal{U}}(\boldsymbol{\theta})$ and used to initialize downstream models during the finetuning on the parallel training set $\mathcal{S}$.

## 3 Cross-breeding: $F_2$-*XEnDec*

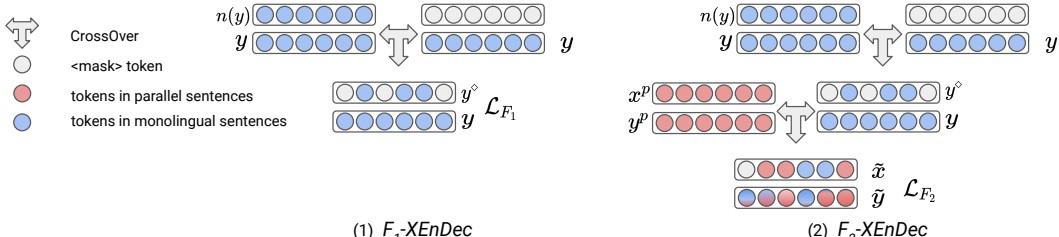

(1) $F_1$-*XEnDec*          (2) $F_2$-*XEnDec*

Figure 1: Illustration of the first filial generation (left) and the second filial generation (right) with *XEnDec*. $F_1$-*XEnDec* is used to generate $(\boldsymbol{y}^\diamond, \boldsymbol{y})$ and produces a self-supervised proxy loss $\mathcal{L}_{F_1}$. $F_2$-*XEnDec* adopts an additional round of *XEnDec* to incorporate parallel data resulting in virtual data $(\tilde{\boldsymbol{x}}, \tilde{\boldsymbol{y}})$ for Equation 3.

For resource-rich translation tasks in which a large parallel corpus $\mathcal{S}$ and (virtually) unlimited monolingual corpora $\mathcal{U}$ are available, our goal is to improve translation performance by exploiting self-supervised signals to complement the supervised learning.

In $F_2$-*XEnDec*, we jointly train NMT models with supervised and self-supervised learning objectives in a single stage. We design a new objective $\mathcal{L}_{F_2}$ and construct virtual data $(\tilde{\boldsymbol{x}}, \tilde{\boldsymbol{y}})$ to bridge the parallel data (in supervised learning) and the pseudo parallel data (in self-supervised learning). The training loss over the virtual data $(\tilde{\boldsymbol{x}}, \tilde{\boldsymbol{y}})$ is computed as:

$$\mathcal{L}_{F_2}(\boldsymbol{\theta}) = \mathbb{E}_{\boldsymbol{y} \in \mathcal{U}} \mathbb{E}_{(\boldsymbol{x}^p, \boldsymbol{y}^p) \in \mathcal{S}}[\ell(f(\tilde{\boldsymbol{x}}, \tilde{\boldsymbol{y}}; \boldsymbol{\theta}), h(\tilde{\boldsymbol{y}}))], \tag{3}$$

where generating $(\tilde{\boldsymbol{x}}, \tilde{\boldsymbol{y}})$ depends on the parallel data $(\boldsymbol{x}^p, \boldsymbol{y}^p)$ and the pseudo parallel data $(\boldsymbol{y}^\diamond, \boldsymbol{y})$.

We propose a method called crossover encoder-decoder (*XEnDec*) that operates on two sentence pairs. As illustrated in Fig. 1, the first generation (Fig. 1(1)) uses *XEnDec* to combine monolingual sentences, thereby incurring a self-supervised proxy loss $\mathcal{L}_{F_1}$ which is equivalent to $\mathcal{L}_{\mathcal{U}}$. The second generation (Fig. 1(2)) applies *XEnDec* between the offspring of the first generation $(\boldsymbol{y}^\diamond, \boldsymbol{y})$ and the parallel sentence $(\boldsymbol{x}^p, \boldsymbol{y}^p)$ to introduce $\mathcal{L}_{F_2}$. The final NMT models are optimized jointly on the original translation loss and the above two auxiliary losses.

$$\mathcal{L}(\boldsymbol{\theta}) = \mathcal{L}_{\mathcal{S}}(\boldsymbol{\theta}) + \mathcal{L}_{F_1}(\boldsymbol{\theta}) + \mathcal{L}_{F_2}(\boldsymbol{\theta}), \tag{4}$$

$\mathcal{L}_{F_2}$ in Equation 4 is used to deeply fuse self-supervised and supervised training at instance level, rather than mixing them across instances mechanically. In the remainder of this section, we first detail *XEnDec*. We then discuss the relation between our approach, pre-training, and adversarial training objectives. Finally, we summarize some important techniques used in our approach and present the overall algorithm.

## 3.1 CROSSOVER ENCODER-DECODER

This section introduces the crossover encoder-decoder (*XEnDec*) – an essential subtask for the proposed method. Different from a conventional encoder-decoder, *XEnDec* takes two training examples as inputs (called parents), shuffles the parents' source sentences and produces a virtual example (called offspring) through a mixture decoder model. Fig. 2 illustrates this process. Formally, let $(\boldsymbol{x}, \boldsymbol{y})$ denote a training example where $\boldsymbol{x} = x_1, \cdots, x_I$ represents a source sentence of $I$ words and $\boldsymbol{y} = y_1, \cdots, y_J$ is the corresponding target sentence. In supervised training, $\boldsymbol{x}$ and $\boldsymbol{y}$ are parallel sentences. As we shall see in Section 3.4, *XEnDec* can be carried out with and without supervision, although we do not distinguish both cases for now.

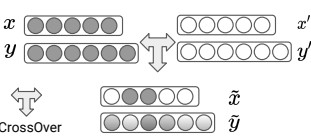

Figure 2: *XEnDec*: CrossOver Encoder-Decoder.

Given a pair of examples $(\boldsymbol{x}, \boldsymbol{y})$ and $(\boldsymbol{x}', \boldsymbol{y}')$ called parents, the crossover encoder shuffles the two source sequences into a new source sentence $\tilde{\boldsymbol{x}}$ calculated from:

$$\tilde{x}_i = m_i x_i + (1 - m_i) x'_i, \tag{5}$$

where $\boldsymbol{m} = m_1, \cdots, m_I \in \{0, 1\}^I$ stands for a series of Bernoulli random variables with each taking the value 1 with probability $p$. If $m_i = 1$, then the $i$-th word in $\boldsymbol{x}$ will be substituted with the word in $\boldsymbol{x}'$ at the same position. For convenience, the lengths of the two sequences are aligned by padding tokens to the end of the shorter sentence. The crossover decoder employs a mixture model to generate the virtual target sentence. The embedding of the decoder's input $\tilde{\boldsymbol{z}}$ is computed as:

$$e(\tilde{z}_j) = \frac{1}{Z} \big[ e(y_{j-1}) \sum_{i=1}^{I} A_{(j-1)i} m_i + e(y'_{j-1}) \sum_{i=1}^{I} A'_{(j-1)i} (1 - m_i) \big], \tag{6}$$

where $e(\cdot)$ is the embedding function. $Z = \sum_{i=1}^{I} A_{(j-1)i} m_i + A'_{(j-1)i} (1 - m_i)$ is the normalization term where $\boldsymbol{A}$ and $\boldsymbol{A}'$ are the alignment matrices for the source sequences $\boldsymbol{x}$ and $\boldsymbol{x}'$, respectively. Equation 6 averages embeddings of $\boldsymbol{y}$ and $\boldsymbol{y}'$ through the latent weights computed by $\boldsymbol{m}$, $\boldsymbol{A}$ and $\boldsymbol{A}'$. The alignment matrix measures the contribution of the source words for generating a specific target word (Och & Ney, 2004; Bahdanau et al., 2015). For example, $A_{j:}$ indicates the contribution scores of each source word for the $j$-th word in the target sentence. For simplicity, this paper uses the attention matrix learned in the NMT model as a noisy alignment matrix (Garg et al., 2019).

Likewise, we compute label vectors for the crossover decoder by:

$$h(\tilde{y}_j) = \frac{1}{Z} \big[ h(y_j) \sum_{i=1}^{I} A_{ji} m_i + h(y'_j) \sum_{i=1}^{I} A'_{ji} (1 - m_i) \big], \tag{7}$$

The $h(\cdot)$ function projects a word onto its label vector, *e.g.* a one-hot vector. The loss of *XEnDec* over $(\boldsymbol{x}, \boldsymbol{y}, \boldsymbol{x}', \boldsymbol{y}')$ is computed as the negative log-likelihood of generating the virtual sentence $\tilde{\boldsymbol{y}}$:

$$\ell(f(\tilde{\boldsymbol{x}}, \tilde{\boldsymbol{y}}; \boldsymbol{\theta}), h(\tilde{\boldsymbol{y}})) = -\log P(\tilde{\boldsymbol{y}} | \tilde{\boldsymbol{x}}; \boldsymbol{\theta}) = \sum_j -\mathbb{E}_{y \in h(\tilde{y}_j)} \log P(y | \tilde{\boldsymbol{z}}_{\leq j}, \boldsymbol{c}_j; \boldsymbol{\theta}), \tag{8}$$

Notice that even though we do not directly observe the "virtual sentences" $\tilde{\boldsymbol{z}}$ and $\tilde{\boldsymbol{y}}$, we are still able to compute the loss using their embeddings. In practice, the length of $\tilde{\boldsymbol{x}}$ is $|\boldsymbol{x}|$ whereas lengths of both $\tilde{\boldsymbol{y}}$ and $\tilde{\boldsymbol{z}}$ are $\max(|\boldsymbol{y}|, |\boldsymbol{y}'|)$.

## 3.2 RELATION WITH OTHER WORKS

Before introducing the proposed method, this subsection shows that the crossover encoder-decoder *XEnDec* (when fed with appropriate inputs) yields learning objectives identical to two recently proposed self-supervised learning approaches, MASS (Song et al., 2019) and BART (Lewis et al.,

Table 1: Comparison with other objectives from the perspective of *XEnDec* with the inputs. Each row shows an input to *XEnDec* and its relation to existing work. $\boldsymbol{y}_{mask}$ is a sentence of length $|\boldsymbol{y}|$ containing only "$\langle mask \rangle$" tokens. $n(\boldsymbol{y})$ is a sentence obtained by corrupting all the words in $\boldsymbol{y}$ through randomly shuffling and dropping. $\boldsymbol{x}_{adv}$ and $\boldsymbol{y}_{adv}$ are adversarial sentences in which all the words are substituted with adversarial words.

| $\boldsymbol{x}$ | $\boldsymbol{y}$ | $\boldsymbol{x}'$ | $\boldsymbol{y}'$ | **Objectives** |
|---|---|---|---|---|
| $\boldsymbol{y}$ | $\boldsymbol{y}$ | $\boldsymbol{y}_{mask}$ | $\boldsymbol{y}$ | MASS (Song et al., 2019) |
| $n(\boldsymbol{y})$ | $\boldsymbol{y}$ | $\boldsymbol{y}_{mask}$ | $\boldsymbol{y}$ | BART (Lewis et al., 2019) |
| $\boldsymbol{x}$ | $\boldsymbol{y}$ | $\boldsymbol{x}_{adv}$ | $\boldsymbol{y}_{adv}$ | Doubly Adversarial (Cheng et al., 2019) |

---

**Algorithm 1:** Proposed $F_2$-XEnDec function.

---

**Input:** Parallel corpus $\mathcal{S}$, Monolingual corpus $\mathcal{U}$, and Shuffling ratios $p_1$ and $p_2$
**Output:** Batch Loss $\mathcal{L}(\boldsymbol{\theta})$.

1 **Function** $F_2\text{-}XEnDec\,(\mathcal{S},\mathcal{U},p_1,p_2)$:
2      **foreach** $(\boldsymbol{x}^p,\boldsymbol{y}^p) \in \mathcal{S}$ **do**
3          Sample a $\boldsymbol{y} \in \mathcal{U}$ with similar length as $\boldsymbol{x}^p$;   // done offline.
4          $(\boldsymbol{y}^{\diamond},\boldsymbol{y}) \leftarrow XEnDec$ over $(n(\boldsymbol{y}),\boldsymbol{y})$ and $(\boldsymbol{y}_{mask},\boldsymbol{y})$ with the shuffling ratio $p_1$;
5          $\mathcal{L}_{\mathcal{S}} \leftarrow$ compute $\ell$ in Equation 1 using $(\boldsymbol{x}^p,\boldsymbol{y}^p)$ and obtain $\boldsymbol{A}$;
6          $\mathcal{L}_{F_1} \leftarrow$ compute $\ell$ in Equation 2 using $(\boldsymbol{y}^{\diamond},\boldsymbol{y})$ and obtain $\boldsymbol{A}'$;
7          $(\tilde{\boldsymbol{x}},\tilde{\boldsymbol{y}}) \leftarrow XEnDec$ over $(\boldsymbol{x}^p,\boldsymbol{y}^p)$ and $(\boldsymbol{y}^{\diamond},\boldsymbol{y})$ with $\boldsymbol{A}$, $\boldsymbol{A}'$ and the shuffling ratio $p_2$;
8          $\mathcal{L}_{F_2} \leftarrow$ compute $\ell$ in Equation 3;
9      **end**
10      **return** $\mathcal{L}(\boldsymbol{\theta}) = \mathcal{L}_{\mathcal{S}}(\boldsymbol{\theta}) + \mathcal{L}_{F_1}(\boldsymbol{\theta}) + \mathcal{L}_{F_2}(\boldsymbol{\theta})$;   // Equation 4.

---

2019), as well as a supervised learning approach called *Doubly Adversarial* (Cheng et al., 2019). Table 1 summarizes the inputs of *XEnDec* to recover the learning objectives of these existing approaches.

*XEnDec* can be used for self-supervised learning. Given arbitrary alignment matrices, if we set $\boldsymbol{x}' = \boldsymbol{y}$, $\boldsymbol{y}' = \boldsymbol{y}$, and $\boldsymbol{x}$ to be a corrupted $\boldsymbol{y}$, then *XEnDec* is equivalent to the denoising autoencoder which is commonly used to pre-train sequence-to-sequence models such as in MASS (Song et al., 2019) and BART (Lewis et al., 2019). In particular, if we allow $\boldsymbol{x}'$ to be a dummy sentence of length $|\boldsymbol{y}|$ containing only "$\langle mask \rangle$" tokens, Equation 8 yields the learning objective defined in the MASS model (Song et al., 2019) except that losses over unmasked words are not counted. Likewise, as shown in Table 1, we can recover BART's objective by setting $\boldsymbol{x} = n(\boldsymbol{y})$ where $n(\cdot)$ is a noise function that alters $\boldsymbol{y}$ by shuffling tokens or dropping them. In both cases, the crossover encoder-decoder is trained with a self-supervised proxy objective to reconstruct the original sentence from one of its corrupted sentences. Conceptually, denoising autoencoder can be regarded as a degenerated *XEn-Dec* in which the inputs are two types of source correspondences for a monolingual sentence, *e.g.*, $n(\boldsymbol{y})$ and $\boldsymbol{y}_{mask}$ for $\boldsymbol{y}$. Even though this connection is trivial, it helps illustrate the power of the *XEnDec* when it is fed with different kinds of input sentences.

*XEnDec* can also be used in supervised learning. We can achieve the translation loss proposed in (Cheng et al., 2019) by letting $\boldsymbol{x}'$ and $\boldsymbol{y}'$ be two "adversarial inputs", $\boldsymbol{x}_{adv}$ and $\boldsymbol{y}_{adv}$, both of which consist of adversarial words at each position. For the construction of $\boldsymbol{x}_{adv}$, we refer to Algorithm 1 in (Cheng et al., 2019). In this case, the crossover encoder-decoder is trained with a supervised objective over parallel sentences.

### 3.3 WHAT DOES $L_{F_2}$ LEARN?

The core term $\mathcal{L}_{F_2}$ introduced in our training objective ( Equation 4) aims to combine parallel and monolingual training examples respectively in supervised and self-supervised learning. Over simply combining $\mathcal{L}_{\mathcal{S}}$ and $\mathcal{L}_{F_1}$ which mechanically mixes the losses across instances, $\mathcal{L}_{F_2}$ further enhances the combination by deeply fusing them at instance level. Let us take an example in Figure 3 to explore what $L_{F_2}$ really learns. $\mathcal{L}_{F_2}$ on the virtual data $(\tilde{\boldsymbol{x}},\tilde{\boldsymbol{y}})$ is optimized by performing the following tasks simultaneously:

$x^p$   Jintian Tianqi feichang hao .       $y^\diamond$ Stock MASK reaches MASK height .

$y^p$   The weather today is very good .      $y$   Stock market reaches new height .

$\tilde{x}$    Stock Tianqi feichang MASK height .

| | 0.85/0.15 | 0.20/0.80 | 0.10/0.90 | 0.75/0.25 | 0.55/0.45 | 0.88/0.12 | 0.05/0.95 |
|---|---|---|---|---|---|---|---|
| $\tilde{y}$ | Stock | market | reaches | new | height | . | PAD |
| | The | weather | today | is | very | good | . |

Figure 3: Illustration of $F_2$-*XEnDec* which is used to incorporate parallel data resulting in virtual data $(\tilde{x}, \tilde{y})$ for Equation 3. "0.85/0.15" indicates the soft combination ratio to generate the virtual data $\tilde{y}$ at each position, *e.g.* , $\sum_{i=1}^{I} A_{0i} m_i / Z = 0.85$ in Equation 7.

1. The NMT model is required to factor the fused source sentences out into two sentences at the target side guided by the alignment.
2. The NMT model is required to predict the tokens corresponding to "MASK" tokens and copy unmasked tokens in the source sentences $y^\diamond$ once it is factored out from $\tilde{x}$. For example, predict "MASK" in $\tilde{x}$ to "new".
3. The NMT model is required to translate words in $x^p$ once it is factored out. For example, translate "Tianqi" to "whether".
4. The NMT model is required to predict tokens that are not picked in the source sentence of the virtual data but can be inferred from the left contextual words. For example, although "market" does not correspond to any word in $\tilde{x}$, it can still be speculated to some extent from the words "Stock", "MASK" and "height". "today" is more likely to be predicted based on "Tianqi" and "feichang".
5. The NMT model is required to separate the loss portion of each above task from the mixed target side.

Integrating all the above five tasks into one task $\mathcal{L}_{F_2}$ via the fused virtual data $(\hat{x}, \hat{y})$ would expect to have the following advantages: (1) each of the above tasks integrated in $\mathcal{L}_{F_2}$ is a harder task than the individually designed task because integration introduces confusions; (2) the virtual points have a more intelligent regularization effect than simply summing up the losses of individual tasks. This is because a virtual data point which deeply fuses the training examples from two specific tasks (self-supervised and supervised tasks) enables the model to try its best to decouple this integration; (3) training the model over the loss $\mathcal{L}_{F_2}$ by integrating the above five tasks performs much more efficiently than simply training five individual tasks.

## 3.4 TRAINING

Algorithm 1 delineates the entire procedure to compute the final loss $\mathcal{L}(\boldsymbol{\theta})$. Specifically, each time we sample a monolingual sentence to avoid the expensive enumeration in Equation 3. To speed up the training, we group sentences by length during offline computation. For the noise function $n(\cdot)$ in Step 4, we follow Lample et al. (2017) and locally shuffle words while keeping the distance between the original and new position at $\leq 3$. There are two techniques to boost the final performance.

**Computing $A$** The alignment matrix $A$ is obtained by averaging the attention weights across all layers and heads. We also add a temperature to control the sharpness of the attention distribution, the reciprocal of which was linearly increased from 0 to 2 during the first $20K$ steps. To avoid overfitting to $A$, we apply dropout to $A$. Notice that we stop back-propagating gradients through $A$.

**Computing $h(\tilde{y})$** Instead of interpolating one-hot labels in Equation 7, we use the prediction vector $f(x, y; \hat{\boldsymbol{\theta}})$ estimated by the model where $\hat{\boldsymbol{\theta}}$ indicates no gradients are back-propagated over it. However, the predictions made at early stages are usually unreliable. We propose to linearly combine the ground-truth one-hot label with the model prediction using a parameter $v$, which is computed as $v f_j(x, y; \hat{\boldsymbol{\theta}}) + (1 - v) h(y_j)$ where $v$ is gradually annealed from 0 to 1 during the first $20K$ steps.

Table 2: Experiments on WMT'14 English-German and WMT'14 English-French translation.

| Models | Methods | En→De | De→En | En→Fr | Fr→En |
|--------|---------|-------|-------|-------|-------|
| Base | Reproduced Transformer | 28.70 | 32.23 | - | - |
| | $F_2$-*XEnDec* | **30.46** | **34.06** | - | - |
| Big | Reproduced Transformer | 29.47 | 33.12 | 43.37 | 39.82 |
| | Ott et al. (2018) | 29.30 | - | 43.20 | - |
| | Cheng et al. (2019) | 30.01 | - | - | - |
| | Yang et al. (2019) | 30.10 | - | 42.30 | |
| | Nguyen et al. (2019) | 30.70 | - | 43.70 | - |
| | Zhu et al. (2020) | 30.75 | - | 43.78 | - |
| | $F_2$-*XEnDec* | **31.60** | **34.94** | **45.15** | **41.60** |

# 4 EXPERIMENTS

## 4.1 SETTINGS

**Datasets** We evaluate our approach on two representative, resource-rich translation datasets, WMT'14 English-German and WMT'14 English-French across four translation directions, English→German (En→De), German→English (De→En), English→French (En→Fr), and French→English (Fr→En). To fairly compare with previous state-of-the-art results on these two tasks, we report the case-sensitive tokenized BLEU scores calculated by the *multi-bleu.perl* script. The English-German and English-French datasets consist of 4.5M and 36M sentence pairs, respectively. The English, German and French monolingual corpora in our experiments come from the WMT'14 translation tasks. We concatenate all the newscrawl07-13 data for English and German, and newscrawl07-14 for French which result in 90M English sentences, 89M German sentences, and 42M French sentences. We use a word piece model (Schuster & Nakajima, 2012) to split tokenized words into sub-word units. For English-German, we build a shared vocabulary of 32K sub-words units. The validation set is newstest2013 and the test set is newstest2014. The vocabulary for the English-French dataset is also jointly split into 44K sub-word units. The concatenation of newstest2012 and newstest2013 is used as the validation set while newstest2014 is the test set.

**Model and Hyperparameters** We implement our approach on top of the Transformer model (Vaswani et al., 2017) using the *Lingvo* toolkit (Shen et al., 2019). The Transformer models follow the original network settings (Vaswani et al., 2017). In particular, the layer normalization is employed after each residual connection rather than before each sub-layer. The dropout ratios are all set to $0.1$ for all the Transformer models except for the Transformer-big model on English-German where $0.3$ is used. We search the hyperparameters using the Transformer-base model on English-German. In our method, the shuffling ratio $p_1$ is set to $0.50$. $p_2$ is sampled from a Beta distribution $Beta(2, 6)$. The dropout ratio for $\boldsymbol{A}$ is $0.2$. For decoding, we use a beam size of $4$ and a length penalty of $0.6$ for English-German and a beam size of $5$ and a length penalty of $1.0$ for English-French. We carry out our experiments on a cluster with $128$ GPUs and update gradients synchronously. The model is optimized with Adam (Kingma & Ba, 2014) following the same learning rate schedule used in (Vaswani et al., 2017), except for *warmup_steps* which is set to $4000$.

## 4.2 MAIN RESULTS

Table 2 shows results comparing our method with several baseline methods on the English-German and English-French datasets across four translation directions. Ott et al. (2018) investigated how to train a strong Transformer model at scale. Our reproduced Transformer model performs comparably with their reported results. Cheng et al. (2019) improved NMT by introducing adversarial attack and defense mechanisms in supervised learning. Nguyen et al. (2019) presented an effective method to boost NMT performance by adopting multiple rounds of back-translated sentences. Both Zhu et al. (2020) and Yang et al. (2019) incorporated the knowledge of pre-trained models into NMT models by treating them as frozen input representations for NMT.

For English-German, our approach achieves significant improvements in both translation directions over the standard Transformer model (up to $+2.13$ BLEU points on the English→German trans-

Table 3: Results on $F_2$-*XEnDec* + Back Translation. English-German is based on the Transformer-base model and English-French is based on the Transformer-big model.

| Methods | En→De | En→Fr |
|---|---|---|
| Transformer | 28.70 | 43.37 |
| Back Translation | 31.38 | 35.90 |
| Edunov et al. (2018) | - | 45.60 |
| $F_2$-*XEnDec* | 30.46 | 45.15 |
| + Back Translation | **32.41** | **46.01** |

Table 4: Results on artificial noisy inputs. "CS": code-switching noise. "DW": drop-words noise.

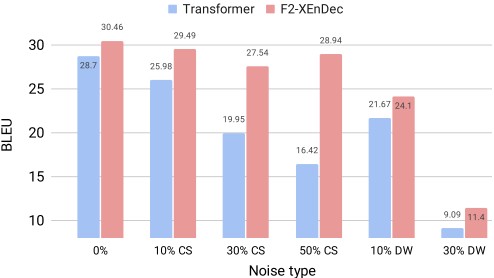

Table 5: Effect of monolingual corpora sizes.

| Methods | Mono. Size | En→De |
|---|---|---|
| | ×0 | 28.70 |
| | ×1 | 29.84 |
| $F_2$-*XEnDec* | ×3 | 30.36 |
| | ×5 | 30.46 |
| | ×10 | 30.22 |

Table 6: Finetuning vs. Joint Training.

| Methods | En→De |
|---|---|
| Transformer | 28.70 |
| + Pretrain + Finetune | 28.77 |
| $F_2$-*XEnDec* (Joint Training) | 30.46 |
| + Pretrain + Finetune | 29.70 |

lation). Even compared with the strongest baseline on English→German, our approach obtains a +0.85 BLEU gain. More importantly, when we apply our approach to a significantly larger dataset, English-French with 36M sentence pairs (vs. English-German with 4.5M sentence pairs), it still yields consistent and notable improvements over the standard Transformer model. The gains over the best baseline (Zhu et al., 2020) on the larger dataset extend to +1.37 BLEU points, which further corroborates the superiority of our approach on resource-rich language pairs.

## 4.3 ANALYSES

**Back Translation as a Noise Function** One widely applicable method to leverage monolingual data in NMT is back translation (Sennrich et al., 2016b). A straightforward way to incorporate back translation into our approach is to treat back-translated corpora as parallel corpora. However, back translation can also be regarded as a noise function $n(\cdot)$ in $F_2$-*XEnDec* (shown in Fig. 1), which can increase the noise diversity. As shown in Table 3, for English→German trained on Transformer-base, our approach yields an additional +1.95 BLEU gain when using back translation to substitute $n(\cdot)$ and also outperforms the back-translation baseline. When applied to the English-French dataset, we achieve a new state of the art result over the best baseline (Edunov et al., 2018). This is interesting because, consistent with what was found in previous works, *e.g.* (Caswell et al., 2019), the standard back translation for English-French hurts the performance of Transformer. These results show that our approach is complementary to the back-translation method and performs more robustly when back-translated corpora are less informative.

**Robustness to Noisy Inputs** Following (Cheng et al., 2019), we validate the robustness of NMT models by word perturbation. Specifically, we design two types of noise to perturb the clean dataset. The first one is code-switching noise (CS) that randomly replaces words in the source sentences with their corresponding target-language words. Alignment matrices are employed to find the target-language words in the target sentences. The other one is drop-words noise (DW) that randomly discards some words in the source sentences. Table 4 shows that our approach exhibits higher robustness than the standard Transformer across all noise types and noise fractions. In particular, our approach performs remarkably stable when exposed to the code-switching noise.

**Effect of Monolingual Corpora Sizes** Table 5 shows the impact of monolingual corpora sizes on the performance for our approach. We find that our approach already yields improvements over the baseline when using a monolingual corpus with size comparable to the bilingual corpus (1x). As we increase the size of the monolingual corpus to 5x, we obtain the best performance with 30.46 BLEU

scores. However, continuing to increase the data size fails to improve the performance any further, and a bigger model with increased capacity might be needed which we leave for future work.

**Finetuning vs. Joint Training** To study the effect of pre-trained models on the Transformer model and our approach, we use Equation 2 to pre-train an NMT model on the entire English and German monolingual corpora. Then we finetune the pre-trained model on the parallel English-German corpus. Models finetuned on pre-trained models usually perform better than models trained from scratch at the early stage of training. However, this advantage gradually vanishes as training progresses (cf. Figure 4 in the Appendix). As shown in Table 6, Transformer with finetuning achieves virtually identical results as a Transformer trained from scratch. Using the pre-trained model over our approach impairs performance. We believe this to be caused by a discrepancy between the pre-trained loss and our joint training loss.

**Ablation Study** Table 7 studies the effect of different components and verifies the design in our approach. When $\mathcal{L}_{F_1}$ is removed, we cannot obtain $\boldsymbol{A}'$ in Algorithm 1 which leads to the failure of calculating $\mathcal{L}_{F_2}$. We design a prior alignment matrix to handle this issue (cf. A.3). We find that removing any component including training techniques (such as dropout $\boldsymbol{A}$ and using model predictions to substitute $h(y_i)$) can result in slightly lower performance. We also tried using Mixup as proposed by Zhang et al. (2018) to replace our second *XEnDec* but observe a large drop compared to $F_2$-*XEnDec*. It is worth noting that just applying *XEnDec* over parallel data (last row) also achieves promising results compared to the Transformer baseline.

Table 7: Ablation study on English-German.

| Different Settings | BLEU |
|---|---|
| Transformer | 28.70 |
| $F_2$-*XEnDec* | 30.46 |
| without $\mathcal{L}_{F_1}$ | 29.55 |
| without $\mathcal{L}_{F_2}$ | 29.21 |
| without dropout $\boldsymbol{A}$ and model predictions | 29.87 |
| without model predictions | 30.24 |
| the second *XEnDec* is replaced by Mixup | 29.67 |
| $\mathcal{L}_S$ with *XEnDec* over parallel data | 29.23 |

## 5 RELATED WORK

The recent past has witnessed an increasing interest in the research community on leveraging pre-training models to boost NMT model performance (Ramachandran et al., 2016; Lample & Conneau, 2019; Song et al., 2019; Lewis et al., 2019; Edunov et al., 2018; Zhu et al., 2020; Yang et al., 2019). Most successes come from low-resource and zero-resource translation tasks. Zhu et al. (2020) and Yang et al. (2019) achieve some promising results on resource-rich translations. They propose to combine NMT model representations and frozen pre-trained representations under the common two-stage framework. The bottleneck of these methods is that these two stages are decoupled and separately learned. Our method, on the other hand, jointly trains self-supervised and supervised NMT models to close the gap between representations learned from either of them with an essential new subtask, *XEnDec*. We also show that our method consistently outperforms previous approaches across several translation benchmarks. Another line of research related to ours is mixing the pixels in labeled image data (Zhang et al., 2018; Yun et al., 2019; Jiang et al., 2020) to enhance generalization and robustness. Our *XEnDec* shares the commonality of combining example pairs. However, *XEnDec*'s focus is on sequence-to-sequence learning for NLP with the aim of using self-supervised learning to complement supervised learning in a joint training framework.

## 6 CONCLUSION

This paper has presented a joint training approach, $F_2$-*XEnDec*, to combine self-supervised and supervised learning in a single stage. The key part is a novel cross encoder-decoder which can be used to "interbreed" monolingual and parallel sentences. Experiments on two resource-rich translation tasks, WMT'14 English-German and WMT'14 English-French, show that our joint training performs favorably against two-stage training approaches and improves the NMT robustness against input perturbations, particularly to code-switching noise. In the future, we plan to further examine the effectiveness of our approach on larger-scale corpora. We also want to design more meaningful noise functions for our approach.

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

## A  APPENDIX

### A.1  TRAINING EFFICIENCY

When training the vanilla Transformer model, each batch contains $4096 \times 128$ tokens of parallel sentences on a 128 P100 GPUs cluster. The training speed is about 0.5 steps/sec on average. As there are three losses included in our training objective ( Equation 4) and the input for each of them are different, we evenly spread the GPU memory budget into these three types of data by letting each batch include $2048 \times 128$ tokens. Thus the total batch size is $2048 \times 128 \times 3$. The training speed on it is about 0.3 steps/sec on average which is about 60% of the standard training speed. The additional computation cost partly comes from implementing the noise function $n(\cdot)$ to construct the corrupted $y^\diamond$, which actually can be reduced by caching noisy data in the data input pipeline. Then the training speed can accelerate to about 0.4 steps/sec.

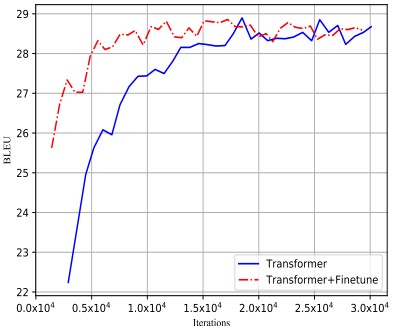 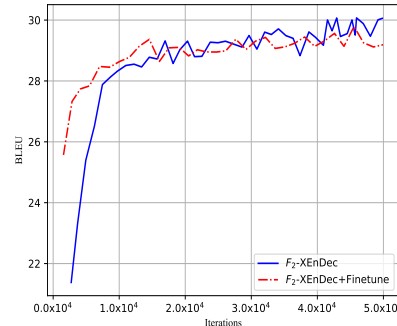

Figure 4: Comparison of finetuning and training from scratch using Transformer and $F_2$-*XEnDec*. In both methods, pre-training leads to faster convergence but fails to improve the final performance after the convergence. The comparison between the figures shows our joint training approach on the left (the blue curve) significantly outperforms against the two-stage training on the right. Final BLEU numbers are reported in Table 6.

## A.2   TRAINING DETAILS

**Data Pre-processing** We mainly follows the pre-processing pipeline (https://github.com/pytorch/fairseq/tree/master/examples/translation) which is also adopted by Ott et al. (2018), Edunov et al. (2018) and Zhu et al. (2020), except for the sub-word tool. To verify the consistency between the word piece model (Schuster & Nakajima, 2012) and the BPE model (Sennrich et al., 2016a), we conduct a comparison experiment to train two standard Transformer models using the same data set processed by the word piece model and the BPE model respectively. The BLEU difference between them is about $\pm 0.2$, which suggests there is no significant difference between them.

**Batching Data** Transformer groups training examples of similar lengths together with a varying batch size for training efficiency (Vaswani et al., 2017). In our approach, when interpolating two source sentences, $\boldsymbol{x}^p$ and $\boldsymbol{y}^\diamond$, it is better if the lengths of $\boldsymbol{x}^p$ and $\boldsymbol{y}^\diamond$ are similar, which can reduce the chance of wasting positions over padding tokens. To this end, in the first round, we search for monolingual sentences with exactly the same length of the source sentence in a parallel sentence pair. After the first traversal of the entire parallel data set, we relax the length difference to 1. This process is repeated by relaxing the constraint until all the parallel data are paired with their own monolingual data.

**Applying Mixup** In Table 7, we tried to substitute the basic operation *XEndec* with Mixup in our approach. When applying Mixup on a pair of training data $(\boldsymbol{x}, \boldsymbol{y})$ and $(\boldsymbol{x}', \boldsymbol{y}')$, Equation 5, Equation 6 and Equation 7 can be replaced by $e(\tilde{x}_i) = \lambda e(x_i) + (1 - \lambda)e(x_i')$, $e(\tilde{z}_j) = \lambda e(y_{j-1}) + (1 - \lambda)e(y_{j-1}')$ and $h(\tilde{y}_j) = \lambda h(y_j) + (1 - \lambda)h(y_j')$, respectively. $\lambda$ is sampled from a Beta distribution. Then we can compute $\mathcal{L}_{F_2}$ accordingly.

## A.3   A PRIOR ALIGNMENT MATRIX

When $\mathcal{L}_{F_1}$ is removed, we can not obtain $\boldsymbol{A}'$ according to Algorithm 1 which leads to the failure of calculating $\mathcal{L}_{F_2}$. Thus we propose a prior alignment to tackle this issue. For simplicity, we set $n(\cdot)$ to be a copy function when doing the first *XEnDec*, which means that we just randomly mask some words in the first round of *XEnDec*. In the second *XEnDec*, we want to combine $(\boldsymbol{x}^p, \boldsymbol{y}^p)$ and $(\boldsymbol{y}^\diamond, \boldsymbol{y})$. The alignment matrix $\boldsymbol{A}'$ for $(\boldsymbol{y}^\diamond, \boldsymbol{y})$ is constructed as follows.

If a word $y_j$ in the target sentence $\boldsymbol{y}$ is picked in the source side which indicates $y_j^\diamond$ is picked and $m_j = 0$, its attention value $A'_{ji}$ if $m_i = 0$ is assigned to $\frac{p}{\|1-\boldsymbol{m}\|_1}$, otherwise it is assigned to $\frac{1-p}{\|\boldsymbol{m}\|_1}$ if $m_i = 1$. Conversely, If a word $y_j$ is not picked which indicates $m_j = 1$, its attention value $A'_{ji}$ is assigned to $\frac{p}{\|\boldsymbol{m}\|_1}$ if $m_i = 0$, otherwise it is $\frac{1-p}{\|1-\boldsymbol{m}\|_1}$ if $m_i = 1$.

