# OpenReview forum: "Self-supervised and Supervised Joint Training for Resource-rich Machine Translation"
_ICLR.cc/2021/Conference — Reject_

### Official Review · AnonReviewer4 · 2020-10-26
**strong empirical results, but where does improvement come from?**

**Rating:** 7
**Confidence:** 4

**Review:**

summary:
--------

This paper introduces a new approach to semi-supervised training of neural machine translation. During training, the traditional supervised loss is complemented with two auxiliary losses: a denoising autoencoder and what the authors call cross-breeding: shuffling together the source side of a sentence pair with an unrelated, noised monolingual sentence, and predicting a virtual target sentence whose embedings and labels are a linear interpolation of the paired target sentence and the monolingual sentence, with interpolation weights being based on an attention matrix of the input words selected during shuffling. Authors report an improvement of around 2 BLEU on WMT14 EN-DE and EN-FR over a purely supervised Transformer, and improvements of 1 BLEU (EN-DE) and 0.4 BLEU (EN-FR) over a system with back-translation. Additionally, the system shows robustness against a type of "code-switching" noise.

strengths:
----------

+ novelty: while there has been some work trying to combine autoencoding and supervised learning for NMT, the idea to create training instances with virtual outputs that combine both tasks is new to my knowledge.

+ strong empirical results: the training objectives compare favourably to previous attempts to use autoencoding as an auxiliary objective for high-resource NMT. It is compatible with back-translation (using back-translations as noise function), and in this setup, obtains a new SOTA on English-French.

+ ablation studies: the paper demonstrates the contribution of the different training objectives, and that joint training is superior to the more commonly used pre-train+fine-tune strategy.

weaknesses:
-----------

- obscure description. The paper makes heavy use of metaphors from biology, describing the approach in terms of "interbreeding" or "marrying" sentences and creating "offspring" over two "filial generations". I found these terms obscuring rather than informative. For example, phrase "we use XEnDec to marry monolingual sentences" is a really roundabout way of expressing that you apply masking to a sentence. Describing a denoising autoencoder as a special case of the crossover encoder-decoder (section 3.1-3.2) also doesn't work for me, because what makes the model special, the creation of a virtual target sentence based on an alignment matrix, isn't applied here.

- robustness results are overclaimed. I was expecting robustness to similar types of noise than in previous work, but the "code-switching" noise employed (randomly replacing source words with the aligned words from the reference translation) is unrealistic and eerily similar to the training scenario (including the reliance on an alignment matrix). There is no strong improvement in the face of "drop-word" noise, and if this noise were of practical relevance, the much simpler strategy of applying word dropout at training time is likely to remedy it.

- open questions about efficiency. Authors use model predictions as decoder inputs (3.3), which prevents parallelisation across time steps at training time. Can authors quantify the training speed difference between the vanilla objective and their combined objective?

- open questions about where improvement comes from. The main novelty is L_F2, which shuffles two inputs and creates a virtual target sentence for prediction. Why this objective is effective is still a bit of a mystery to me, and I would have liked more insights into this. Most pressingly, using the predictions as decoder inputs reminds me of work on scheduled sampling, and could have a substantial effect that may be unrelated to self-supervised learning. How much does using the prediction vector as input embedding contribute to the quality? (Ablation studies indicate that dropout on A and using the predictions together contributes 0.6 BLEU, which is about half of the benefit of L_F2) Does the vanilla objective improve when using model predictions instead of (or in addition to) gold inputs?

- related to the above question (where does improvement come from?), back-translation. Best results are reported when using back-translation instead of a denoising autoencoder. It's unclear how this result fits into the narrative. If authors still consider back-translation "self-supervised learning", then this weakens the contribution that "we show that self-supervised learning is complementary to supervised learning" (which is well-known for back-translation already), and also doesn't fit the narrative in the introduction that self-supervised learning "provides a much weaker learning signal that is easily dominated by the signal coming from supervised learning". If back-translation experiments are seen as separate from the self-supervised learning experiments, it is misleading to imply that self-supervised learning achieves a new state of the art.

recommendation:
---------------

I'm currently borderline on this paper. The main strength of the paper are the strong empirical results, but I have misgivings about some of the central claims of the paper, most importantly the robustness results and positioning the paper as a way of strengthening the self-supervised learning signal, when the reason for the performance improvement is still obscure.

additional questions to the authors and minor points:
-------------

- some implementation details were unclear to me. When using back-translation as a noise function n(·), do you still apply masking in addition? If so, is there a good rationale why masking would only be helpful for back-translated data, but not for real parallel data? I also don't understand what you mean with "applying XEnDec over parallel data". Are you mixing two sentence pairs (shuffling the source sentences together and predicting an interpolated target sentence)?

- it was not made clear until 3.3 that L_F1 is the same as L_U.

- equation 7: the second sum should be \sum_{i=1}^I

- I note that English-German results with back-translation are not directly comparable to Edunov et al. (2018) because they are based on Transformer base. Why did you not report results with Transformer big?

UPDATE
-----------

the author response has addressed some concerns (efficiency concerns; unclarities about some ablations studies; reason for not comparing to Edunov for EN-DE) well, and I have slightly increased my rating.

I maintain that the description is needlessly obscure in places, even in the revised version (while it is technically correct that a denoising autoencoder could be described as a crossover encoder where the two parent outputs are identical, this is needlessly convoluted), and a major concern about the framing of the paper was not resolved in the response (if back-translation is self-supervised learning, then some of the claims about limitations of self-supervised learning are untrue, and the findings lose novelty. If back-translation isn't self-supervised learning, then it's misleading to imply that self-supervised learning led to a new SOTA).

---

> ### Author Response · Authors · 2020-11-12
> **Part 1 To AnonReviewer4 (1 of 2)**
>
> Q1: Obscure description.
>
> A1: Your point is well taken and we acknowledge that, although the current description of the method describes the basic procedure of forming training data points, it needs to illustrate more details such as virtual data points creation with alignment matrix. In a new version, we plan to (1) have a better illustration of the process. (2) introduce the motivation of the method and rationale why it works--for now, please refer to answer A2 to AnonReviewer1. About section 3.2, we believe that our description is correct although it is a little unclear to you. For example, when combining (n(y), y) and (y_{mask}, y), because their target sentences are identical, the combination has not been affected by a specific alignment. Thus given arbitrary alignment matrices, the combination of $y$ in ($n(y)$, $y$) and $y$ in ($y_{mask}$, $y$) is still $y$. We will make it more clear. In addition, we clarify that our results and improvement still hold and thank you for your suggestion.
>
> Q2: Robustness results are overclaimed.
>
> A2: We apologize for this and will revise it to improve the robustness to noisy text, particularly to code-switching noise. To construct a noisy “code-switching” test set, the resorted alignment matrix is generated by the standard Transformer model rather than our $F_2$-XEndec model, which favors the baseline Transformer in principle. Our approach is robust against “code-switching” noise partly because of our $L_{F_{2}}$ loss which combines a source-language sentence and a target-language sentence into a virtual sentence. This type of noise usually appears on the social media platform where users post code-switching content. We think our approach can be applicable to this translation scenario. The improvement for drop-word noise is not as strong as code-switching noise  as you pointed out although our approach consistently performs better than the standard Transformer over the drop-word noise data set across different ratios. We believe that the robustness against drop-word noise can be improved if we add the drop-word operation in the noise function n($\cdot$) in our approach.
>
> Q3: Open questions about efficiency.
>
> A3: About the training cost, each batch in the standard Transformer contains 4096x128 tokens of parallel sentences and the training speed on a 128-GPUs cluster is about 0.52 steps/sec on average. In our approach, to load our approach into GPU memory, we reduce the batch size to 2048x128 tokens of parallel sentences and 2048x128 tokens of monolingual sentences. As our training losses also includes $L_{F_{2}}$ which computes loss over 2048x128 tokens of virtual sentence pairs  (refer to equation (4)), the total batch size is 2048x128x3. The training speed is about 0.30 steps/sec on average which is about 60% of the standard training speed. This is because we have extra computation overhead to construct noisy sentence pairs by executing noise function $n(\cdot)$ , which actually can be reduced by caching noisy data in the input pipeline.  However, at the same iteration, our results are consistently higher than the baseline methods. For the issue of using model predictions as decoder inputs (3.3) which prevents parallelisation across time steps at training time, actually it is used in equation 7 to compute label vectors rather than in equation 6 to compute the decoder input (see “Compute $h(\tilde{y})$” in Section 3.3). We can also use one-hot labels in equation 7 instead of predictions to reduce the computation cost on this point. But it can slightly hurt the best result reported in the paper, dropping to 30.24 from 30.46.
>
> Q4: Comparison to schedule sampling.
>
> A4: Thanks for your insight that using the predictions as decoder inputs like scheduled sampling. In schedule sampling, prediction vectors are dynamically generated step by step and formulated as one-hot vectors. The prediction estimated by the last step is fed into the current step. Referring to “Compute $h(\tilde{y})$” in Section 3.3, prediction vector is only used to calculate equation 7 for label vectors. When computing decoder input in equation 6, we do not use prediction vectors but their original one-hot embeddings. We will make it more clear in the revised paper.
>
> Q5: Best results are reported when using back-translation instead of a denoising autoencoder. It's unclear how this result fits into the narrative.
>
> A5: We believe that back-translation is a more effective noise function n($\cdot$) for NMT which can be incorporated into our $F_{2}$-XEndec framework (refer to Fig 1.). Compared to some common and simple noise functions such as shuffle, deletion and masking, back-translation can introduce more natural and diverse noises that can be not easy to simulate using several simple noise functions. This experiment also shows that our approach is compatible with other data augmentation methods including back-translation.

---

> > ### Author Response · Authors · 2020-11-12
> > **Part 2 To AnonReviewer4 (2 of 2)**
> >
> > additional questions
> >
> > Thanks for your insightful comments and we will revise our paper accordingly.
> >
> > Q6: Some implementation details were unclear to me.
> >
> > A6: Sorry for the confusion. In the “Back Translation as a Noise Function” section, back translation can also be regarded as a noise function n(·) in F$_2$-XEnDec indicating that we go through the entire procedure in the right sub-figure of Figure 1. In the first generation XEndec, back-translation sentences are applied with masking words and the second XEndec combines it with a parallel sentence pair.
> >
> >  "applying XEnDec over parallel data" means that we just perform a single XEndec over two parallel sentences. Thus no masking is applied. We will make it more clear in the revised paper.
> >
> >
> > Q7: I note that English-German results with back-translation are not directly comparable to Edunov et al. (2018) because they are based on a Transformer base. Why did you not report results with Transformer big?
> >
> > A7: In Table 3, we compare with Edunov et al. (2018) over WMT’14 English-French rather than English-German. In this setting, the data set including parallel data and monolingual data, the network architecture, the data preprocessing pipeline and hyperparameters for the Transformer big model nearly agree between ours and Edunov et al. (2018)  to ensure a fair comparison. In addition, all the experiments in Edunov et al. (2018) are based on the Transformer big (refer to Section 4.2 in their paper). We do not compare to them on English-German because their results are based on WMT'18 dataset.

---

> ### Author Response · Authors · 2020-11-24
> **Response to UPDATE**
>
> Thanks for your helpful updates. We have revised the claim about back translation in the abstract and introduction of the latest version.  We also make our paper more readable by clarifying the connection between denoising autoencoders and our crossover encoder. We will continue to streamline it.

---

### Official Review · AnonReviewer1 · 2020-10-27
**interesting and impressive performances**

**Rating:** 5
**Confidence:** 5

**Review:**

Summary:

This submission proposes a joint training of self-supervised training and supervised training for neural machine translation (NMT), especially for the rich-resource language datasets. The method proposed, F$_2$-XEnDec, exploits the "crossover" operation of the monolingual sentences and bilingual data pairs in the encoder-decoder framework, and train the self-supervised (on monolingual data) objective and supervised (on bilingual data) objective, and the mixed version (crossover) together, without a clear per-train and fine-tuning stage like previous BERT related works. Experiments are conducted on two rich-resource language pairs: WMT14 English-German and WMT14 English-French translation tasks. The results set a new state-of-the-art record for English-French, and superior performances are achieved on English-German language pairs.  The authors also analyze the robustness and ablation studies of their method.

Overall comments:

The method proposed in this submission is interesting, the crossover operation conducted on the monolingual dataset is similar to the previous works like MASS and BART, but different in the bilingual sentences mixed version. The authors give clear descriptions and comparisons with previous methods. Another difference is that previous works only adopt pre-train and fine-tune stages separately, while this method is to jointly train the self-supervised and supervised objectives, on both monolingual data and bilingual dataset. The crossover operation of the source sentence is natural to be straight forward, while it is interesting for the target side sentences, both on monolingual and target sentences. The most impressive part is the experimental results, the authors give very strong performances on two widely-acknowledged benchmark datasets. The results are very competitive and demonstrate the method is quite effective.

Detailed comments:

There are many good points of this submission, as I talked about in the last paragraph. Besides, the writing is clear and easy to follow. The overall presentation is high-quality in terms of writing, e.g., the formulations, the figures. Therefore, I would like to turn to point out some concerns from my side, and I still want to know more about this work, please see the follows:

* The first point is about the crossover operation performed on the second stage, the monolingual data and the bilingual data. The source sentences ($x$, $x'$) are directly mixed together through a mask operation to be a new source sentence, and the target sentences are mixed through a weighted version, as the equation (6) and (7) show. This releases some questions to me: (1) the source sentence $x^p$ and the noised source sentence $y^{\diamondsuit}$ is combined through equation (5), but this mix the space from both source and target, and also interrupts the meaningful space. This augmentation operation is different from the previous works, therefore, how to guarantee this operation is reasonable and effective. It seems more explanations need to be shown. (2) Similarly, as for the target sentences, the mixed version of the monolingual target $y$ and bilingual $y^\{\diamondsuit}$ is most probably a non-sense sentence. Therefore, it is hard to think of the reason or the motivation for such mixing, especially in a weighted version as equation (7) and (6). More explanations and interpretations are needed for those operations. (3) Of course, the operation is similar to the mixup operations proposed previously, but the relationship is not clearly discussed in this submission. Mixup is the most related work since the method in this paper, or jointly training method can be regarded as a data augmentation method with two loss objectives from both monolingual and bilingual sets. Therefore, I feel okay about the story presented and motivated by the pre-train methods in this work, but I do feel the most related work is mixup and related data augmentation methods for NMT.
* Specifically for the crossover operation, besides the relations and explanations for the mixed operation, the weighted label vectors are computed in an attention-based approach by leveraging the attention matrices. As $A$ and $A'$ are not good at the first training steps, the authors add the trick of linearly increasing temperature for them. Hence, the related question is to see the performance of label vector without the weighted version, with a similar operation as equation (5). The advantage of this weighted label vector compared to others need to be discussed since this is more complex than other methods, the training cost is also increased. The importance of the normalization factor $Z$, the necessity of current formulation. More motivation for this modeling is expected.
* As for the training tricks, for computing $h$, I do not clearly understand why predicted $f$ should be added to the target labels. Is it related to scheduled sampling, or what else? Why the label vector is formulated in such a way?
* Since the authors compared with other pre-train methods, MASS, BART, therefore, the experimental results are expected to make a clear comparison.
* For the details of the experiments, the authors use $128$ GPUs to run for the experiments, which is a huge number that is not easy for others to reproduce. The authors are encouraged to open-source the code and settings. A small question is for the dropout values, is attention dropout and relu dropout adopted? It seems to be not clear. The training cost should also be discussed since the algorithm contains several dependencies in the three objectives.
* In analyses, the authors see the effect of monolingual corpora size for English-German task, which contains $4.5M$ bilingual datasets, therefore, the monolingual size of $3\times$, $5\times$ and $10\times$ are less than $89M$ and $90M$ as used in the experimental settings, but the results $30.46$ is from $3\times$, is it correct? If it is like this, the authors should give more descriptions of this point to give people a clear data setting. One another small question is about the mixup replacement mentioned in ablation study, can you give more words on how you perform mixup in this setting, $x=x_1 + x_2$ corresponds to what operations for two sentences with several words. Is it similar to the masked replacement as equation (5)?

In a word, I want to hear more about the details of this submission and I am willing to increase the score if questions are answered.

----post-update----

Hi, I thank the authors to give useful feedback about several concerns. But I am still questioned about the cross-sentence operation, even the authors give an example of the different tasks, which is hard to convince me for such kind of learning can achieve such results. The 5 tasks as the author described are much harder, which somehow indicates that bilingual data is not so necessary for machine translation (but it should be aware, this is not unsupervised machine translation, the training spirit is totally different).  Random data augmentation could be very very strong. The authors are expected to give a code implementation and the trained model to give a more convincing result.

---

> ### Author Response · Authors · 2020-11-12
> **Part 1 To AnonReviewer1 (1 of 2)**
>
> Q1: The source sentence and the noised source sentence are combined through equation (5), but this mixes the space from both source and target, and also interrupts the meaningful space.
>
> Q2: The motivation for the crossover operation over the target sentences.
>
> A2: I reply to these two questions together about the motivation of our F$_{2}$-XEndec using an example.
>
> A monolingual example:
> src. Stock MASK reaches MASK high.
> target: Stock market reaches new high.
>
> A bilingual example:
> src: JinTian TianQi FeiChang Hao
> target: The weather today is very good.
>
> The virtual data point:
> src: Stock TianQi FeiChang MASK high
> target: implicitly combined in embeddings.
>
> Training NMT on the virtual data points means performing the following tasks simultaneously:
>
> (1) Language/source separation: NMT model needs to factor the mixed two sentences out into two sentences, guided by the mixed target side, therefore, mixed loss.
>
> (2) The model needs to predict the MASKed tokens, once the source is factored out of the mixed source.
>
> (3) The model needs to translate Chinese once Chinese is separated out.
>
> (4) The model needs to predict tokens that are not picked in the source sentence of the virtual data but can be inferred from the left contextual words.
>
> (5) The model needs to separate the loss portion of each above task from the mixed target side.
>
> Integrating all the above 5 via fused virtual data points would expect to have the following advantages:
>
> (1) Each of the above tasks is a harder task than the individually designed task because integration/mixing introduces confusions.
>
> (2)  Furthermore, the virtual points have a more intelligent regularization effect than just adding the individual loss of individual tasks, because, compared to merely adding the loss up, a virtual data point deeply fuses/integrates the example & loss (separation) and intends to make the model work hard to decouple this integration.
>
> In these senses, whether the data points being mixed are from the same meaning space or not may not matter much, because our goal is to let the NMT model  decouple them and learn, which follows the spirit of self-learning.
>
> This would interfere with the self attention, but this could also make the model learn stronger attention that can be more discriminative, through the virtual points.
>
> Q3: Relation to Mixup.
>
> A3: Our work is inspired by mixup (Zhang et al. 2017) and cutmix (Yun et al. 2019) which generates virtual images in the vicinity of two observed images. Our XEndec also tries to produce virtual sentence pairs by combining two observed sentence pairs under a sequence-to-sequence learning framework. In particular, the second generation F2-XEndec is applied to use virtual sentence pairs constructed over parallel sentence pairs and monolingual sentence pairs to help self-supervised learning to complement supervised learning.  In the ablation study as shown in Table 7, we replace XEndec with a simple mixup technique and find it performs much worse than our XEndec.
>
> Q4: To see the performance of label vector without the weighted version.
>
> A4: When we want to construct label vectors with a 0-1 version in equation (5) rather than with the weighted version in equation (7), it is not easy as we do not know which $m_{j}$ in the target sentence should be 1 or 0 given a virtual source sentence $\tilde{x}$. If we just randomly sample $m_{j}$ to be 1 or 0 for each target position, the performance becomes a lot worse. Thus a more precise method is to sample $m_{j}$ according to their respective accumulated contribution scores, $\sum_{i}^{I}A_{ji}m_{i}$ and  $\sum_{i}^{I}A_{ji}^{\prime}(1 - m_{i})$. Our latest experiment (not included in the paper) shows that if we construct the decoder’s input in Eq.(6) by using this method to sample $m_{j}$, our best result on WMT’14 En->De is 30.30 (drop from 30.46). And when this method is applied to $h(y)$ in equation 7, there is also no significant improvement. The normalization Z aims to make the sum of  $\sum_{i}^{I}A_{ji}m_{i}$ and  $\sum_{i}^{I}A_{ji}^{\prime}(1 - m_{i}$) be 1, which ensures $h(\tilde{y})$ to be a well-formed label vector in that $\sum_{e\in h(\tilde{y})} e = 1$. The intuition behind using alignment to guide the combination of two target sentences is that if source words of a sentence pair appear most in a virtual source sentence, their corresponding target words recognized by an alignment matrix should take up more space in the virtual target sentence

---

> > ### Author Response · Authors · 2020-11-12
> > **Part 2 To AnonReviewer1 (2 of 2)**
> >
> > Q5: As for the training tricks, for computing, I do not clearly understand why predicted  should be added to the target labels.
> >
> > A5:  Instead of interpolating one-hot labels in equation (7), we use the prediction vector $f(x, y; \theta)$ estimated by the model. Actually in Transformer, the label-smoothing technique is applied to one-hot labels. MLE to train the Transformer aims to optimize the KL divergence between $f(x, y; \theta)$ and one-hot labels. In our approach, we think the model prediction vector is a more informative label compared to a one-hot label. If we do not use $f$, the best BLEU score decreases to 30.24 from 30.46. We will add this result to the ablation study in a future version. In schedule sampling, prediction vectors are dynamically generated step by step and formulated as one-hot vectors. The prediction estimated by the last step is fed into the current step. Referring to “Compute
> > ” in Section 3.3, prediction vector is only used to calculate equation 7 for label vectors. When computing decoder input in equation 6, we do not use prediction vectors but their original one-hot embeddings. We will make it more clear in the revised paper.
> >
> > Q6： Since the authors compared with other pre-train methods, MASS, BART, therefore, the experimental results are expected to make a clear comparison.
> >
> > A6: $L_{F_{1}}$ in our approach can be replaced by arbitrary pre-train objectives,  MASS or BART. Direct comparisons with theirs are less meaningful because they focus more on learning representations.
> > The $L_{F_{1}}$ used in our training is similar to the BART which learns a denoised autoencoder while the noise functions are slightly different. In Table 7, we remove $L_{F_{2}}$ and only combine self-supervised learning and supervised learning objectives to obtain much lower performance than incorporating $L_{F_{2}}$. In our latest experiment, we try to design more meaningful noise functions and achieve better results with 30.73 on English-German (vs. 30.46 in the paper).
> >
> > Q7: For the details of the experiments, the authors use 128 GPUs to run for the experiments, which is a huge number that is not easy for others to reproduce.
> >
> > A7: Our Transformer model is implemented using the Lingvo toolkit following the transforme_vaswani_big settings in the fairseq (https://github.com/pytorch/fairseq/blob/master/fairseq/models/transformer.py). Either attention dropout or relu dropout is not turned on in our model. The fairseq toolkit supports using an 8-GPUs machine to simulate a 128-GPUs cluster by accumulating gradients. Our baseline methods such as Zhu et al. (2020) use this simulation to train a Transformer model.  We will release our network setting and all the details. More details about the training cost, each batch in the standard Transformer contains 4096x128 tokens of parallel sentences and the training speed is about 0.52 steps/sec on average. In our approach, to load our approach into GPU memory, we reduce the batch size to 2048x128 tokens of parallel sentences and 2048x128 tokens of monolingual sentences. As our training losses also includes $L_{F_{2}}$ which computes loss over 2048x128 tokens of virtual sentence pairs  (refer to equation (4)), the total batch size is 2048x128x3. The training speed is about 0.30 steps/sec on average which is about 60% of the standard training speed. This is because we have extra computation overhead to  construct noisy sentence pairs by executing noise function $n(\cdot)$ , which actually can be reduced by caching noisy data in the input pipeline. However, at the same iteration, our results are consistently higher than the baseline methods (Reproduced Transformer and Back-translation).
> >
> >
> >
> > Q8: More details about monolingual data size and mixup operation.
> >
> > A8: 30.46 is obtained by using 5x monolingual data according to Table 5, which are sampled from 89M German sentences.  We will detail these experimental data settings and the mixup operation in the revised paper as you suggested.
> >
> > Given two sentence pairs (x, y) and (x’, y’), we sample a $\lambda$ value from a Beta distribution.
> > $e(\tilde{x}_{i}) = \lambda e(x_i) + (1-\lambda) e(x^{\prime}_i)$ for equation 5,
> >
> > $e(\tilde{z}_{j}) = \lambda e(y_j) + (1 - \lambda) e(y^{\prime}_j)$ for equation 6 (the subscript of y should be j-1, but it can not show normally in the system),
> >
> > $e(h(\tilde{y}_{j})) =  \lambda h(y_j) + (1-\lambda) h(y^{\prime}_j)$ for equation 7.

---

### Official Review · AnonReviewer3 · 2020-10-28
**An interesting method that remains to be fairly evaluated**

**Rating:** 5
**Confidence:** 5

**Review:**

This paper presents a method, inspired from genetics algorithms, to trained jointly self-supervised and supervised NMT. The claims are this new method and state-of-the-art BLEU scores for En-Fr WMT14


Strengths:
- the method inspired from genetics is original and interesting
- the method improves over vanilla Transformer
- the analysis is interesting by pointing out the key parameters of the proposed method


Weaknesses
- the main weakness of this paper is by far its evaluation: the comparisons of BLEU scores presented in Table 2 and 3 are meaningless. For instance, Zhu et al. (2020) uses sacreBLEU while this paper use multi-bleu. On En-Fr, it is well-known that the differences between these different BLEU may be within 5 BLEU points (!). It is then impossible to assess whether this work is really better than previous work from these results. sacreBLEU must be reported instead of, or in addition to, multi-bleu scores. Moreover, the work compared in Table 2 used different training data and different pre-processing, the difference in BLEU may then only come from these data differences and not from the frameworks. I actually expect this method to be worse than Zhu et al. (2020) considering that their sacreBLEU scores is less than 2 BLEU points below the multi-bleu scores reported in this paper. The second half of the paper has to be rewritten almost entirely to make a meaningful evaluation of this approach.
- the motivation for using cross-over encoder-decode is difficult to understand, probably because it is only done very briefly in Section 1 (I would recommend to make separate section for motivation)

Questions:
- Section 4.1: how reporting tokenized BLEU scores is fair while previous work, mentioned in Table 2, report sacreBLEU scores? You probably misread Section 5.1 from Zhu et al. (2020). For Nguyen et al. (2019), this is unclear what are the reported BLEU scores, but for xx-to-English it seems to be also sacreBLEU scores. Ott et al (2018) also report sacreBLEU scores in addition to multi-bleu. I think you have all you need to replace the scores in Table 2 by sacreBLEU scores. It would dramatically improve the paper.
- Some of the methods compared in Table 2 uses different sets of training data and preprocessing, maybe I misunderstood something, but even with comparable BLEU scores, how can we assess that this is the proposed method that is better and not the training data or preprocessing? To make a scientifically valid conclusion I think there is no other way than reproducing previous work. There is no need to reproduce all the work from Table 2.  I think the most relevant one to reproduce is probably the work of Zhu et al. (2020).

---

> ### Comment · AnonReviewer4 · 2020-11-11
> **actually, authors' comparison is mostly fair**
>
> I was asked by the AC whether your observations about SacreBLEU change my assessment. Actually, I disagree with this criticism.
>
> R3 says that Ott et al. (2018), Zhu et al. (2020), and Nguyen et al. (2019) use SacreBLEU, but it's a bit more complicated than that:
>
> It's true that Ott et al. (2018) report both multi-bleu.perl and SacreBLEU, but this paper correctly compares against the multi-bleu.perl results ("WMT only").
>
> Nguyen et al. (2019)'s comment about using SacreBLEU only applies to the low-resource datasets. See appendix C, where they say "We measure the performance in standard tokenized BLEU." for WMT.
>
> For Zhu et al. (2020): "We use multi-bleu.perl to evaluate IWSLT’14 En↔De and WMT translation tasks for fair comparison with previous work. For the remaining tasks, we use a more advance implementation of BLEU score, sacreBLEU for evaluation" (Zhu et al., 2020, p. 6). Since the paper under review only looks at WMT, they're correctly comparing results with multi-bleu.perl.
>
> Yes, tokenized BLEU is less trustworthy because we can't be sure about the tokenization, and I would encourage people to report SacreBLEU in addition to tokenized BLEU whenever possible, but given the limitation that previous work mostly reported only tokenized BLEU, the comparison by the authors looks fair in this respect.

---

> ### Author Response · Authors · 2020-11-11
> **Comparison is fair**
>
> Q1:  how reporting tokenized BLEU scores is fair while previous work, mentioned in Table 2, report sacreBLEU scores?
>
> A1: According to the last paragraph in Section 5.1 from Zhu et al. (2020), they “use multi-bleu.perl to evaluate IWSLT’14 En↔De and WMT translation tasks for fair comparison with previous work. “. I think the BLEU scores on WMT translation tasks including WMT’14 En->Fr and WMT’14 En->De in their paper are also calculated by “multi-bleu script” with tokenized BLEU scores. Most of previous work conducting experiments on WMT’14 En->Fr and En->De also report tokenized BLEU scores. To fairly compare to theirs, we use multi-bleu.perl to compute scores. For Nguyen et al. (2019), we believe that they report tokenized BLEU scores too from the results in Table 2 of their paper. For example, their baseline results on WMT’14 En->De is 28.4 for Transformer and 29.3 for scale Transformer which are consistent with tokenized BLEU scores reported by Vaswani et al. (2017) and Ott et al. (2018). Among all the baseline methods, only Ott et al. (2018) and Edunov et al. (2018) report both tokenized BLEU and sacreBLEU.
>
> Q2: Some of the methods compared in Table 2 use different sets of training data and preprocessing.
>
> A2: For the parallel training data, we mainly follow fairseq toolkit (https://github.com/pytorch/ fairseq/tree/master/examples/translation), which is also adopted by Zhu et al. 2020,  to use almost exactly the same processing pipeline except for the script to split words into sub-word units. In our paper, we adopt the word piece model (Schuster & Nakajima, 2012), which is basically equivalent to BPE (Sennrich et al., 2016). We also report our standard Transformer baseline results in Table 1: 29.47 vs. 29.30 on En->De and 43.37 vs. 43.20  on En->Fr between our reproduced Transformer and Ott et al. (2018) verify that ours are  consistent with Ott et al. (2018) in terms of data and pre-processing.  Zhu et al. (2020) also achieve similar results on these same datasets. Therefore, all methods compared use equivalent data and pre-processing.

---

> > ### Comment · AnonReviewer3 · 2020-11-11
> > **more fair than I thought**
> >
> > Thank you for your comment.
> > I acknowledge that I misread Zhu et al. (2020). They also used multi-bleu and it makes your comparisons of BLEU scores more realistic than I thought. I will increase my rating consequently.
> >
> > However, this is still not right. I do not know how we can assess from Table 2 whether one system is significantly better than the others. 10 years ago, when it was not normal to compare scores from different paper, reviewers would have requested for statistical significance testing.
> > At least for better future work, you should report on sacreBLEU scores.
> >
> > I  am not convinced by A2. There are too many approximations. Small differences in data and pre-processing can lead to different BLEU scores and change the ranking of the systems in Table 2.
> >
> > A good paper in MT (and Science in general) should reproduce previous work with its own data to be 100% sure that the proposed method is better, and not the data. I am not asking to reproduce all the works in Table 2. Selecting one good baseline is enough to be convincing.

---

> ### Author Response · Authors · 2020-11-17
> **To  AnonReviewer3**
>
> Q3: the motivation for using cross-over encoder-decode is difficult to understand, probably because it is only done very briefly in Section 1 (I would recommend to make separate section for motivation)
>
> A3; Thanks for your suggestions. I am very sorry to forget replying to this question. Please See answer A2 to AnonReviewer1 for more rationale of our method. We will revise our paper.
>
> Continued A2:
> 1. Both Zhu et al. (2020) and we reproduced the standard Transformer achieving similar results on WMT'14 En-De and En-Fr (refer to Table 2 in our paper and Table 3 in their paper).
> 2. Our improvements over Zhu et al. (2020) are +0.85 and +1.37 respectively on WMT'14 En-De and En-Fr, which are significant improvements.
> 3. We use the same pre-processing pipeline including the important tokenization script by following fairseq (https://github.com/pytorch/ fairseq/tree/master/examples/translation), except for the BPE tool. Referring to A.1 in Zhu et al. (2020), they also use the same script.
> 4. To verify the consistency between the word piece model and BPE model, we conduct new experiments to train two standard Transformer models using the same data set processed by the word piece model and BPE model respectively. The BLEU difference between them are about ±0.2.

---

### Official Review · AnonReviewer2 · 2020-10-29

**Rating:** 5
**Confidence:** 4

**Review:**

This paper proposes a joint training strategy that combines supervised learning on parallel data and self-supervised learning on monolingual data for NMT. The monolingual sentences are corrupted with word order shuffling and masking. And a cross encoder-decoder is introduced to fuse the parallel source-side sentence and the corrupted monolingual sentence. The NMT models are optimized jointly with cross-entropy loss of parallel data, reconstruction loss for monolingual data, and cross-entropy of the virtual fused data.

The proposed method is simple and straightforward, with seemingly good results for rich-resource languages; can be easily reproduce in any existing toolkits.

Main results (table 2) should compare with BT and noised BT (Edunov et al. 2018) using the same amount of monolingual data as well. Current comparisons with previous BT/pre-training approaches in the main results (table 2) are not exactly fair, since lots of them are not leveraging large-scale monolingual data of all available newscrawl combined. (correct me if I'm wrong).

I'm also a little skeptical about the fusion of source-side sentence x and target-side sentence y* into the input sequence \tilde x. Would this potentially hampers self-attention in the encoder or the enc-dec attention? Empirically, the last row of table 3 (+BT) shows that replacing n(y) with BT sentence x' leads to significantly better performance, which might be caused by this?
This also leads to the question, what is the best fusing scheme (i.e. input (x, y), (x', y')) for the proposed method? Is this the most data efficient way of combing parallel and mono data compared with previous objectives (e.g. noisy BT, DAE, MASS, etc.)?

How is the training and convergence speed for the proposed method, compared with standard Transformer, with BT/noised BT with same amount of mono data?

---

> ### Author Response · Authors · 2020-11-12
> **To AnonReviewer2**
>
> Q1: Main results (Table 2) should compare with BT and noised BT (Edunov et al. 2018) using the same amount of monolingual data as well.
>
> A1: For WMT’14 English-French, Edunov et al. (2018) use all available French newscrawl data in the WMT translation task, like our work. BT results (Back Translation in Table 3) are reproduced by us so that the size of monolingual data is kept the same.
>
> Q2: I'm also a little skeptical about the fusion of source-side sentence x and target-side sentence y* into the input sequence \tilde x. Would this potentially hamper self-attention in the encoder or the enc-dec attention?
>
> A2: The Transformer model with high model capacity is capable of fitting massive training examples. We expect such virtual examples to instead strengthen the performance. Significant improvements on the clean test set in Table 2 and 3 demonstrate the generalizability of our approach. See answer A2 to AnonReviewer1 for more rationale.
>
> Q3:Empirically, the last row of table 3 (+BT) shows that replacing n(y) with BT sentence x' leads to significantly better performance, which might be caused by this? This also leads to the question, what is the best fusing scheme (i.e. input (x, y), (x', y')) for the proposed method?
>
> A3: In Table 3, we use back translation to replace the noise function n($\cdot$) in $F_{2}$-XEndec showing that back translation is a more effective noise function and back translation data can also be combined with parallel data using our proposed XEndec. Table 7 shows some of the possibilities to combine different types of data. For example, “$L_{s}$  with XEnDec over parallel data” means that we apply XEndec over two parallel sentences, more specifically, both (x, y) and (x', y') are parallel sentences. When $L_{F_{1}}$ is removed, we directly combine parallel sentence pairs and monolingual sentence pairs without being guided by alignment matrics. Comparisons among these types of combinations show that using our approach over the combination of parallel data and back translation data performs best.
>
> Q4:  Is this the most data efficient way of combining parallel and mono data compared with previous objectives (e.g. noisy BT, DAE, MASS, etc.)?
>
> A4: In terms of efficiency, please see A5. We expect that the combination of parallel and back translation data to be a more effective method as the back translation can be regarded as a more effective and diverse noise function with natural transformations to the input data compared to simple noise functions such as shuffle and masking. The result in Table 3 demonstrates its effectiveness.
>
>
>
> Q5:How is the training and convergence speed for the proposed method, compared with standard Transformer, with BT/noised BT with same amount of mono data?
>
> A5: About the training cost, each batch used by the standard Transformer contains 4096x128 tokens of parallel sentences and the training speed on a 128-GPUs cluster is about 0.5 steps/sec on average. Our approach tries to evenly spread the GPU memory budget into the three types of data (loss), each having 2048x128 tokens in training data batch (Eq. 4). The total batch size is 2048x128x3. The training speed is about 0.3 steps/sec on average which is about 60% of the standard training speed. The additional computation cost partly comes from constructing noisy sentence pairs by executing noise function $n(\cdot)$, which actually can be reduced by caching noisy data in the input pipeline. For the back translation, it requires to train a backward translation model and decode the monolingual data which takes about twice as long as training a standard Transformer.
>
> However, at the same iteration, our results are consistently higher than the baseline methods (Reproduced Transformer and Back Translation).

---

### Author Response · Authors · 2020-11-20
**Paper Revision**

Dear reviewers,

Thanks for your constructive comments. We uploaded a revised version where the major changes are highlighted with the bleu color.

1. Use a specific example to introduce the motivation more clearly.
2. Give more details about the efficiency of our approach.
3. Some modifications in writing.

Please feel free to let us know whether there are additional suggestions.

---

> ### Author Response · Authors · 2020-11-25
> **New rebuttal version**
>
> In order to satisfy the length limit of the rebuttal version,  we put some of our changes in the appendix section. Please pay some attention to them.

---

### Decision · Program_Chairs · 2021-01-07
**Final Decision**

**Decision:**

Reject

**Comment:**

in this submission, the authors propose a sophisticated pretraining strategy for neural machine translation based on the paradigm of self-supervised learning. despite some interesting and potentially significant improvement in various machine translation settings, the reviewers as well as i myself could not determine where specifically those improvements come from. is it their particular strategy of pretraining or is it just self-supervised learning in general? in order for this question, which i believe is a key question to be answered, more thorough ablative experiments and/or comparison to other self-supervised learning based pretraining algorithms, such as MASS & BART which were discussed as similar and motivational in the submission, must be done. when these are done, the submission will be much stronger and attract much more interest.